

# Functional response of *Neoseiulus californicus* (Acari: Phytoseiidae) to *Tetranychus urticae* (Acari: Tetranychidae) at different temperatures

Maryam Mumtaz[1,*], Vattakandy Jasin Rahman[2,*], Tahseen Saba[3], Tingting Huang[1], Yuxin Zhang[1], Chunxian Jiang[1] and Qing Li[4]

[1] College of Agronomy, Department of Agricultural Entomology, Sichuan Agricultural University, Chegndu, Sichuan, China
[2] Department of Zoology, TKM College of Arts and Science, Kollam, Kerala, India
[3] College of Forestry, Sichuan Agricultural University, Chengdu, Sichuan, China
[4] College of Agronomy, Department of Agricultural Entomology, Sichuan Agricultural University, Chengdu, Sichuan, China
* These authors contributed equally to this work.

Corresponding author
Qing Li, liq8633@163.com

## ABSTRACT

Environmental factors like temperature have a great impact on the predation potential of biological control agents. In the present study, the functional response of the predatory mite *Neoseiulus californicus* (Acari: Phytoseiidae) to the pest mite *Tetranychus urticae* (Acari: Tetranychidae) at moderate to high temperatures under laboratory conditions was determined. The study aimed to understand the prey-predator interaction under different temperatures and prey densities. Five constant temperatures (24 °C, 27 °C, 30 °C, 33 °C, and 36 °C), and thirteen prey densities (4, 5, 8, 10, 12, 15, 16, 20, 24, 25, 30, 32, and 40) of each stage (adult, nymph, larvae, and egg stage) were employed in the experiment. Observations were made 24 h after the start of each experiment. Results revealed that the predatory mites showed type II functional response to adult females of *T. urticae*, whereas type I to other stages (nymphs, larvae, and eggs) of *T. urticae*. The predation capability of adult predatory mites on *T. urticae* was significant at 24–36 °C. The instantaneous attack rate ($a$) of *N. californicus* increased and the handling time ($Th$) decreased with an increase in temperature. The maximum attack rate was recorded at 36 °C (1.28) for the egg stage. The longest handling time was (0.78) for the larval stage of *T. urticae* at 30 °C. Daily consumption increased with increasing prey density. Maximum daily consumption was observed at 33 °C (30.00) at the prey density of 40. Searching efficiency decreased with the increase in prey density but was found to increase with the rise in temperature. *N. californicus* was found to be voracious on the larval and egg stages. Conclusively, the incorporation of *N. californicus* at earlier stages (larvae and eggs) of *T. urticae* would be beneficial under warm conditions because managing a pest at its initial stage will save the crop from major losses. The results presented in this study at various temperatures will be helpful in different areas with different temperature extremes. The results of the functional response can also be applied to mass rearing, quality testing, and integrated pest management programmes.

## INTRODUCTION

The two-spotted spider mite, *Tetranychus urticae* Koch (Acari: Tetranychidae), is a very devastating agricultural pest to 140 families of plants and found to feed on more than 1,100 plant species in which 150 species of plants are of economic value (*Pavela, 2017*). It poses a major threat to vegetables, large trees, horticultural plants, and deciduous fruit trees (*Oliveira et al., 2007*; *Migeon, Nouguier & Dorkeld, 2010*). It is present all over the world in various agro-ecosystems and can withstand a wide range of temperatures (*Takafuji, 1994*; *Sabelis, 1985*). Large-scale uses of pesticides not only resulted in the development of pesticide resistance in this mite but also in the eradication of the majority of the spider mite's natural enemies, thereby decreasing the predation pressure on these mites (*Wu et al., 2016*). Additionally, due to its high net reproduction and rapid developmental rates, *T. urticae* becomes resistant to acaricides over time (*Uddin et al., 2017*). Considering the development of pesticide resistance, the use of biological control is thought to be a crucial control measure against *T. urticae* across the globe (*Amoah et al., 2016*; *Gigon, Camps & Le Corff, 2016*; *Seiedy, Soleymani & Hakimitabar, 2017*).

Natural enemies are used in some agricultural systems as an alternative to chemical pest management. Numerous elements, including temperature, which is crucial for greenhouse crops, have an impact on the effectiveness of pest mites' natural enemies (*Jafarian & Jafari, 2016*; *Tello-Mercado, Zarzar-Maza & Suarez-Pantoja, 2017*; *Shakarami & Bazgir, 2017*; *Huyen et al., 2017*; *Jensen et al., 2017*; *Toldi et al., 2017*; *Stavrinides & Mills, 2011*). The rapid spread of these pests in both indoor and outdoor conditions has been controlled using natural enemies of pest mites. Many studies on the effects of acarophagous ladybirds (*Mori et al., 2005*), predatory mites (*Castagnoli & Simoni, 1999*; *Lester, Thistlewood & Harmsen, 2000*), and predatory anthocorids (*Gitonga et al., 2002*) on the population of pest mites (*Kishimoto, 2003*; *Gotoh et al., 2004*) have shed promising light on the biocontrol measures.

*Neoseiulus californicus* McGregor is a globally distributed predatory mite from the family Phytoseiidae. It has been found on many deciduous trees and crops in Asia, America, and Europe. It was sold in many nations around the world due to its effective biological control potential against pest mites, such as tarsonemids and spider mites (*Copping, 2001*). In biological control programmes, it is vital to evaluate the effectiveness of natural enemies before releasing them. The analysis of foraging behavior, particularly their functional responses, is one of the crucial techniques for determining the effectiveness of these natural enemies (*Fathipour et al., 2006*). Functional response refers to the number of successfully attacked prey per predator as a function of prey density. It demonstrates how a natural enemy reacts to a changing prey density. Predator and prey species, pesticides, host plant variety, predator age, and both biotic and abiotic factors are some of the elements that determine the type of functional response (*Pakyari, Kasirloo & Arbab, 2016*). Three categories of functional responses are described by *Holling (1959a, 1959b, 1961)*. In Type I,

the prey intake and reproductive rates rise steadily until the curve reaches a maximum plateau. Type II describes a cyrtoid curve that rises at a declining rate until it reaches a plateau. Type III describes a sigmoid curve which accelerates quickly till it reaches an inflection point and then slows down before plateauing.

Predator-prey interactions are pervasive, regulate the flow of energy up the trophic levels, and significantly affect the constitution of ecosystems. They are commonly evaluated using the functional response, which describes the relationship between a predator's rate of foraging and the availability of food. The functional response is a key factor in the abundance, diversity, and dynamics of ecological communities. The functional response also represents all of the behaviours, characteristics, and tactics that prey and predators utilize to avoid each other. It is a crucial factor in the future evolution of both predators and prey, as well as a clear reflection of evolutionary history. Even though functional responses are crucial, there have been surprisingly few attempts to summarise or even briefly evaluate them. Studying the functional response of predatory mites to varying prey densities and temperature conditions is essential for understanding their impact on pest populations. This research will help refine ecological models and support integrated pest management strategies that rely on these mites as natural biological control agents. Moreover, it will enhance the understanding of food web dynamics and the ecology of predatory mites, which play a crucial role in regulating herbivorous pest populations. By incorporating these insights into ecological models, one can predict how environmental changes, such as temperature fluctuations, may affect predator-prey interactions and overall ecosystem functioning. Ultimately, this knowledge will contribute to the development of effective conservation strategies and sustainable agricultural practices that utilize predatory mites to promote biodiversity and ecosystem health while reducing reliance on chemical pesticides.

This study aimed to quantify the effect of temperature on the functional response of *N. californicus*. We hope this study would offer a better understanding of the prey-predator interaction and help develop an effective strategy to incorporate *N. californicus* for the biological control of pest mites. The results of the study conducted at moderate to high temperatures will find application while incorporating *N. californicus* under areas with varying extreme temperatures. The altering prey density and extreme temperatures will give researchers a clear understanding of how *N. californicus* changes its behaviour and predatory capabilities under various prey availability and environmental conditions. Thus the objectives of the present study were to evaluate the functional response, searching efficiency, and consumption rate of *N. californicus* to each prey stage of *T. urticae* at moderate to high temperatures under laboratory conditions.

## MATERIALS AND METHODS

### Laboratory rearing of *T. urticae* and *N. californicus*

Both *T. urticae* and *N. californicus* were collected first to start the stock colony. *N. californicus* was collected from Pujiang (Chengdu, China) while, *T. urticae* was collected from Wenjiang (Chengdu, China). Adult two-spotted spider mites were transferred to healthy bean (*Canavalia gladiata* Dc.) leaves and placed on a moist cotton pad kept inside

Petri dishes. Water was added to the rearing unit in order to keep the cotton moist and avoid the leaves from drying out. The colony was maintained by cutting the deteriorated leaves with red spider mites and putting them dorsal sides down on fresh leaves. Gravid females of *N. californicus* were collected from the red spider mite-infested fields. These females were reared on infested bean leaves placed on moist cotton pad kept in Petri dishes. Cotton pads were regularly kept damp to prevent the egress of predatory mites. Every 2 days, the Petri dishes were checked to see if the predator overpopulated, and any leaves that were deficient in red spider mites were swapped out for fresh bean leaves. Both spider mites and predatory mites were kept alive in a lab environment with $25 \pm 1\,°C$, $75 \pm 5\%$ RH and 16:8 (L:D) photoperiod. The temperature was maintained at a particular unit by placing the experimental setup in an incubator and monitored at times.

**Functional response and daily consumption at different temperatures**

Adult females of *N. californicus* were transferred from the stock culture to bean leaves having prey mites. These adult female mites were allowed to lay eggs for 12 h and then were removed after egg-laying, and the eggs were left alone to finish the postembryonic development. Six-day-old females were separated and starved for 24 h, and a single predatory mite was used in each treatment. The following densities of different stages of *T. urticae* were supplied from the stock culture with a camel hair brush for the treatment at each temperature (24 °C, 27 °C, 30 °C, 33 °C, and 36 °C): 4, 5, 8, 10, 12, 15, 16, 20, 24, 25 adults, 8, 10, 15, 16, 20, 24, 25, 30, 32, 40 Nymphs, 8, 10, 15, 16, 20, 24, 25, 30, 32, 40 larvae and 8, 10, 15, 16, 20, 24, 25, 30, 32, 40 eggs. Each treatment was replicated five times. The numbers of prey consumed by the predatory mite were recorded after 24 h. We also evaluated the searching efficiency of predatory mites.

**Data analysis**

The functional responses were identified by fitting the data to the Holling disc equation (*Holling, 1959a*), $Na/P = aNT/(1 + aThN)$; where $Na$ is the number of successful attack per predator ($P$) during a given time period ($T$) which in this case is 1 day; $N$ is the initial density of the prey and $a$ and $Th$ are, respectively, the successful attack rate and the time required by predator to handle the prey. According to *Holling (1963)*, handling time is the time that the predator requires to pursue, kill and digest its prey. The parameters $a$ and $Th$ were calculated using linear regression technique where in $1/Na$ was regressed on $1/N$. $a$ is the reciprocal of the slope and $Th$ is the intercept. $a/Th$ value indicates the effectiveness of predation. Maximum predation rate, $K$ was calculated as $T/Th$.

The data on the daily consumption of prey were analyzed with one-way ANOVA using SPSS software. Estimation of searching efficiency depends on the population density of the prey. During this experiment, searching efficiency was estimated by using the following equation:

$$S = a/1 + aThN_0$$

where, S is searching efficiency: $N_0$ is the initial number of prey: $a$ is the attack rate, and $Th$ is the time required to dispose of a prey.

**Table 1 Functional response of *N. californicus* adult females to the different life stages of *T. urticae* at different temperatures.**

| Temperature | Stage | *a* (Attack rate) | *Th* (Handling time) | *a/Th* | *T/Th* ( *K*) | *Na* (Functional response equation) | *R²* (Correlation coefficient) |
|---|---|---|---|---|---|---|---|
| 24 °C | Adult | 0.357 | 0.037 | 9.581 | 26.81 | 0.357*N*/(1 + 0.013*N*) | 0.861 |
| | Nymph | 0.609 | 0.013 | 46.82 | 76.92 | 0.609*N*/(1 + 0.008*N*) | 0.936 |
| | Larvae | 0.537 | 0.021 | 25.09 | 46.02 | 0.537*N*/(1 + 0.011*N*) | 0.946 |
| | Egg | 0.626 | 0.005 | 127.7 | 204.1 | 0.626*N*/(1 + 0.003*N*) | 0.934 |
| 27 °C | Adult | 0.412 | 0.020 | 20.59 | 50.00 | 0.412*N*/(1 + 0.008*N*) | 0.928 |
| | Nymph | 0.749 | 0.014 | 52.06 | 69.44 | 0.749*N*/(1 + 0.011*N*) | 0.953 |
| | Larvae | 0.772 | 0.012 | 66.00 | 85.47 | 0.772*N*/(1 + 0.009*N*) | 0.974 |
| | Egg | 0.606 | 0.003 | 195.6 | 322.6 | 0.606*N*/(1 + 0.002*N*) | 0.959 |
| 30 °C | Adult | 0.539 | 0.016 | 34.09 | 63.29 | 0.539*N*/(1 + 0.009*N*) | 0.971 |
| | Nymph | 0.653 | 0.009 | 72.59 | 111.1 | 0.653*N*/(1 + 0.005*N*) | 0.985 |
| | Larvae | 0.614 | 0.011 | 55.32 | 90.09 | 0.614*N*/(1 + 0.007*N*) | 0.988 |
| | Egg | 0.673 | 0.002 | 320.7 | 476.2 | 0.673*N*/(1 + 0.001*N*) | 0.968 |
| 33 °C | Adult | 0.541 | 0.005 | 112.9 | 208.3 | 0.541*N*/(1 + 0.003*N*) | 0.989 |
| | Nymph | 0.666 | 0.004 | 185.0 | 277.7 | 0.666*N*/(1 + 0.002*N*) | 0.992 |
| | Larvae | 0.734 | 0.004 | 183.5 | 257.0 | 0.734*N*/(1 + 0.003*N*) | 0.996 |
| | Egg | 0.624 | 0.012 | 46.96 | 80.00 | 0.624*N*/(1 + 0.008*N*) | 0.994 |
| 36 °C | Adult | 0.560 | 0.003 | 186.7 | 333.3 | 0.560*N*/(1 + 0.002*N*) | 0.955 |
| | Nymph | 0.614 | 0.012 | 50.78 | 82.64 | 0.614*N*/(1 + 0.007*N*) | 0.986 |
| | Larvae | 0.705 | 0.001 | 587.7 | 833.3 | 0.705*N*/(1 + 0.001*N*) | 0.954 |
| | Egg | 0.588 | 0.012 | 47.01 | 80.00 | 0.588*N*/(1 + 0.007*N*) | 0.985 |

Note:

*Na*, number of prey consumed; *N*, initial prey density; *Th*, handling time (day); *a*, attack rate; *K*, maximum predation rate; *R²*, coefficient of determination.

## RESULTS

### Functional response

A type II functional response was shown by *N. californicus* to adult females whereas type I to other stages of *T. urticae* and it was fitted to the Holling's disc equation (Table 1). The results on Table 1 showed that successful attack rate (*a*) significantly increased as the temperature rose from 24 °C and 36 °C with a levelling off at temperatures above 27 °C, while the handling time (*Th*) decreased. The longest handling time was recorded at 24 °C (0.04) and the shortest at 36 °C (0.001). The handling time varied at different feeding stages.

Mean attack rate and handling time also showed how the attack rate increased and handling time first increased and then decreased (Fig. 1). At various temperatures, correlation coefficient values ranged from 0.86 to 0.99. Maximum predation (*K*) and *a/Th* values were recorded at the egg stage at 24 °C to 30 °C, followed by larval or nymphal stages in an irregular pattern. Based on these values, *N. californicus* preferred eggs the least at 33 °C and 36 °C. *N. californicus* preferred nymphs of *T. urticae* most, followed by larvae and adults at 33 °C, but larvae, followed by adults and nymphs at 36 °C.

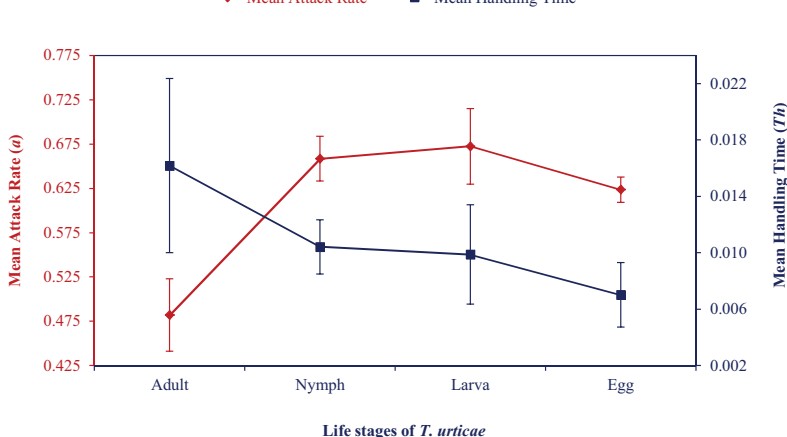

**Figure 1 Mean attack rate (*a*) and handling time (*Th*) of *N. californicus* at different stages of *T. urticae*.**

## Daily consumption

Daily consumption of *N. californicus* at various temperatures and different prey densities was analyzed with ANOVA. Maximum consumption was recorded at 33 °C (33.00) at the prey density of 40. The highest consumption was recorded in the larval stage followed by egg, nymph, and adult (Fig. 2). Predators consumed more prey as the temperature increased but showed a levelling off above 30 °C (Table 2).

## Searching efficiency

Temperature also had a different impact on *N. californicus*'s ability to find pest mites at various stages of pest mites. The searching efficiency dropped at the each temperature as the prey density increased. The highest searching efficiency was observed at 30 °C and 33 °C for most of the prey stages. Searching efficiency on the larval and egg stages of prey mites was higher showing that *N. californicus* prefers these stages than others (Fig. 3).

## DISCUSSION

Results of the present study show the effect of moderate to high temperatures as well as fluctuating prey densities on the functional response of *N. californicus*. Various prey densities were subjected to study in order to see how the predatory mite's behaviour altered in tandem with the rapidly shifting prey densities. The performance of *N. californicus* was found to increase as temperature increased but showed a levelling off beyond 30 °C. The attack rate was found to be highest on larval stage of the prey and lowest on adult stage. The longest handling time was recorded at 24 °C (0.04) and the shortest at 36 °C (0.001). The functional response of *N. californicus* to two-spotted spider mite *T. urticae* at moderate to high temperature ranges has only been the subject of a very small number of investigations. This study paid attention to all four stages of *T. urticae* since all stages of spider-mite have dominant roles in crop damage and economic loss.

In this investigation *N. californicus* female showed Type II functional response to adult females and type I to other stages of *T. urticae* at five different temperatures. Phytoseiid

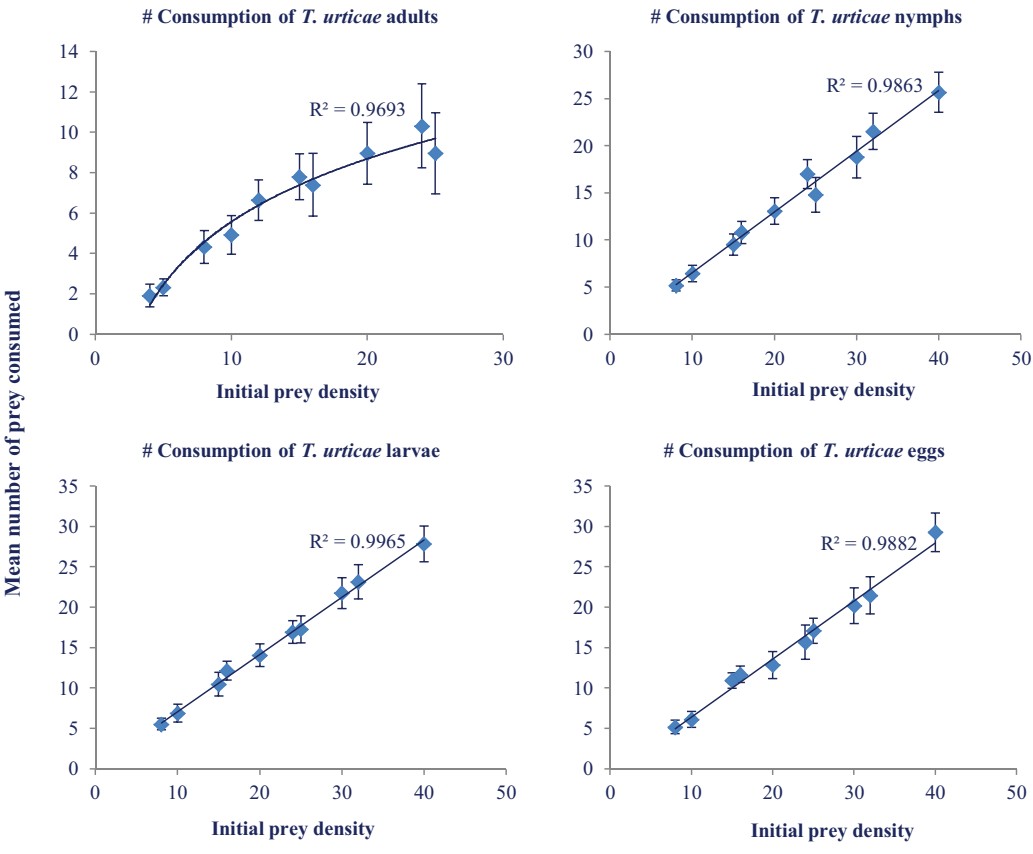

**Figure 2 Mean number of prey consumed (±SE) by adult females of *N. californicus* at different densities of various life stages of *T. urticae*.**

mites frequently exhibit the Type II functional response, which has been observed in *N. californicus* (*Ahn, Kim & Lee, 2010*; *Kuştutan & Cakmak, 2009*; *Döker & Kazak, 2016*; *Merlin et al., 2022*) as well as other phytoseiid species such as *A. longispinosus* (Evans) (*Zhang et al., 1998*), *Phytoseiulus macropilis* (Banks) (*Poletti, Maia & Omoto, 2007*), *Chileseius camposi* Gonzalez & Schuster (*Sepúlveda & Carrillo, 2008*), *Galendromus occidentalis* (Nesbitt) (*Xiao & Fadamiro, 2010*), *Kampimodromus aberrans* (Oudemans) (*Kasap & Atlihan, 2011*), *Iphiseius degenerans* (Berlese) (*Fantinou et al., 2012*), *N. longispinosus* (*Rahman et al., 2012*), *Phytoseiulus persimilis* Athias-Henriot (*Seiedy et al., 2012*), *Galendromus flumenis* (*Ganjisaffar & Perring, 2015*), *Cydnoseius negevi* (*Alatawi et al., 2018*), and *Euseius concordis* (*da Silveira et al., 2020*). If the consumption rate increases linearly with the quantity of food up to a certain threshold at which it stabilises, the functional response is said to be type I. Examples of this kind of functional response are rare; they are usually observed in interactions between herbivores and plants and some predator-prey relationships between invertebrates. In this study *N. californicus* female showed type I functional response to nymph, larva and egg stages of *T. urticae*. A type I functional response is characterized by a linear increase in predation rate with increasing prey density until saturation occurs. In this study, it means that *N. californicus*'s predation rate increases linearly with the increasing density of *T. urticae* (4, 5, 8, 10, 12, 15,

**Table 2 Daily consumption of *N. californicus* on different pest densities of *T. urticae* under five temperatures.**

| Stages | Treatments | Density of *T. urticae* | | | | | | | | | | | | |
|---|---|---|---|---|---|---|---|---|---|---|---|---|---|---|
| | | 4 | 5 | 8 | 10 | 12 | 15 | 16 | 20 | 24 | 25 | 30 | 32 | 40 |
| Adult | 24 °C | 1.40 ± 0.24b | 1.40 ± 0.24ab | 4.00 ± 0.71ab | 3.80 ± 0.86ab | 5.20 ± 1.28ab | 3.40 ± 1.03ab | 3.20 ± 1.32ab | 5.40 ± 1.36ab | 6.60 ± 1.60a | 6.80 ± 2.39a | – | – | – |
| | 27 °C | 1.60 ± 0.40d | 2.40 ± 0.51d | 3.80 ± 1.07bcd | 3.20 ± 0.86cd | 8.20 ± 0.86ab | 11.00 ± 0.71a | 9.00 ± 1.52a | 7.20 ± 2.46abc | 10.60 ± 2.75a | 8.40 ± 1.56a | – | – | – |
| | 30 °C | 2.00 ± 0.45d | 2.80 ± 0.37cd | 4.00 ± 0.71cd | 5.40 ± 1.03cd | 5.60 ± 1.07cd | 5.40 ± 1.72cd | 7.40 ± 1.63bc | 12.80 ± 1.43a | 11.00 ± 2.68ab | 11.20 ± 2.51ab | – | – | – |
| | 33 °C | 2.20 ± 0.58f | 2.60 ± 0.51ef | 5.00 ± 0.71def | 6.20 ± 0.86cde | 6.20 ± 0.86cde | 8.80 ± 1.28bcd | 9.00 ± 1.30bc | 10.40 ± 1.21ab | 12.60 ± 2.06ab | 14.00 ± 1.92a | – | – | – |
| | 36 °C | 2.40 ± 0.51e | 2.40 ± 0.51e | 4.80 ± 0.86de | 6.00 ± 1.14cde | 8.00 ± 1.00bcd | 10.40 ± 0.92b | 8.40 ± 2.06bcd | 9.00 ± 1.22bc | 10.80 ± 1.35b | 4.40 ± 1.63a | – | – | – |
| Nymph | 24 °C | – | – | 4.40 ± 0.51d | 5.80 ± 1.07d | – | 9.00 ± 1.41cd | 9.80 ± 1.28bcd | 9.20 ± 1.85bcd | 15.80 ± 1.74ab | 9.40 ± 2.46bcd | 14.60 ± 3.32abc | 14.80 ± 2.59abc | 20.60 ± 2.80a |
| | 27 °C | – | – | 6.00 ± 0.71e | 6.20 ± 0.86e | – | 9.00 ± 1.52de | 9.40 ± 1.43de | 14.40 ± 1.43bcd | 14.60 ± 2.42bcd | 13.60 ± 2.20cd | 15.60 ± 3.00bc | 19.60 ± 2.01b | 25.40 ± 1.77a |
| | 30 °C | – | – | 5.40 ± 0.51e | 6.40 ± 0.93de | – | 9.00 ± 1.00cde | 11.20 ± 1.35cd | 12.00 ± 1.61c | 18.00 ± 1.30b | 17.80 ± 1.39b | 18.20 ± 2.42b | 23.40 ± 2.42a | 26.60 ± 1.96a |
| | 33 °C | – | – | 5.40 ± 0.51g | 7.00 ± 0.71g | – | 9.60 ± 1.08ef | 11.00 ± 1.00de | 14.60 ± 1.08cd | 18.80 ± 1.07b | 17.00 ± 1.41bc | 22.80 ± 1.07a | 24.40 ± 1.70a | 25.20 ± 2.27a |
| | 36 °C | – | – | 5.00 ± 0.71f | 6.80 ± 0.86f | – | 11.00 ± 0.71e | 12.80 ± 0.86de | 15.20 ± 1.07cd | 17.80 ± 1.15c | 16.20 ± 1.88cd | 22.80 ± 1.24b | 25.40 ± 0.93b | 30.60 ± 1.78a |
| Larva | 24 °C | – | – | 4.20 ± 1.02f | 7.00 ± 1.30ef | – | 10.60 ± 1.69de | 13.20 ± 1.07cd | 15.00 ± 1.30cd | 17.00 ± 1.58bc | 18.00 ± 1.58bc | 21.40 ± 2.50b | 21.60 ± 2.44b | 30.40 ± 1.63a |
| | 27 °C | – | – | 5.80 ± 0.66f | 7.60 ± 0.75ef | – | 9.20 ± 1.49def | 9.80 ± 1.35cdef | 12.40 ± 1.86cde | 14.80 ± 2.13bcd | 15.80 ± 2.44abc | 19.80 ± 1.77ab | 23.00 ± 2.70a | 23.80 ± 3.33a |
| | 30 °C | – | – | 5.60 ± 0.51f | 6.40 ± 1.29f | – | 9.60 ± 1.63ef | 12.20 ± 1.24de | 13.60 ± 1.29cde | 15.80 ± 1.28cd | 17.40 ± 1.03c | 22.00 ± 1.52b | 24.00 ± 1.30b | 28.40 ± 1.86a |
| | 33 °C | – | – | 6.00 ± 0.70g | 7.40 ± 1.08g | – | 11.80 ± 0.86f | 13.40 ± 1.21ef | 15.40 ± 1.21de | 18.00 ± 1.00cd | 19.40 ± 1.50c | 24.20 ± 1.24b | 25.40 ± 1.40b | 33.00 ± 1.41a |
| | 36 °C | – | – | 6.00 ± 0.71d | 6.0 ± 1.00d | – | 11.20 ± 1.07cd | 12.00 ± 1.00c | 14.00 ± 1.41bc | 19.00 ± 1.00ab | 15.80 ± 1.85bc | 21.40 ± 2.50a | 21.80 ± 2.73ab | 23.60 ± 2.80a |
| Egg | 24 °C | – | – | 5.00 ± 0.94d | 5.40 ± 1.29d | – | 11.60 ± 1.36bcd | 11.00 ± 0.71bcd | 8.60 ± 2.01cd | 13.00 ± 3.14bc | 13.00 ± 2.91bc | 18.20 ± 2.88b | 14.80 ± 3.33bc | 28.00 ± 2.34a |
| | 27 °C | – | – | 5.00 ± 0.71d | 5.60 ± 0.51d | – | 10.20 ± 0.86cd | 12.40 ± 0.93c | 13.60 ± 1.50c | 13.40 ± 2.93c | 14.40 ± 1.29c | 21.00 ± 2.51b | 16.40 ± 3.20bc | 28.80 ± 2.54a |
| | 30 °C | – | – | 5.80 ± 0.86d | 6.20 ± 0.86d | – | 10.20 ± 0.86cd | 11.60 ± 1.21c | 12.00 ± 2.17c | 19.60 ± 1.03b | 18.80 ± 1.49b | 18.00 ± 2.21b | 25.60 ± 1.75a | 30.20 ± 2.58a |
| | 33 °C | – | – | 5.20 ± 0.86f | 6.80 ± 1.39f | – | 11.40 ± 0.93e | 11.60 ± 1.17e | 16.20 ± 1.24d | 17.60 ± 1.72d | 19.60 ± 1.21cd | 23.20 ± 1.28bc | 27.00 ± 1.58ab | 29.80 ± 2.08a |
| | 36 °C | – | – | 4.80 ± 0.86d | 6.40 ± 0.93d | – | 11.20 ± 0.86c | 11.80 ± 1.07c | 13.80 ± 1.43c | 14.80 ± 1.71c | 19.60 ± 0.93b | 20.60 ± 2.16b | 23.60 ± 1.78b | 29.60 ± 2.32a |

**Note:**
Data in the table are mean ± SE. Data in the same group followed by different letters indicate significant difference at $P \leq 0.05$ Level (Duncan's new multiple range test).

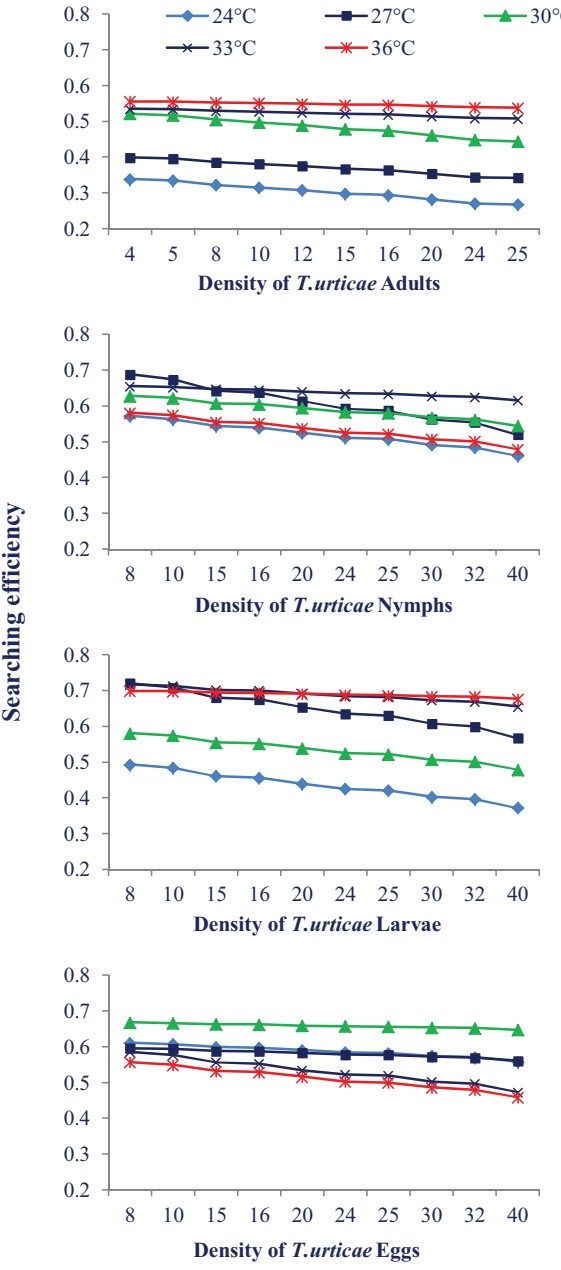

**Figure 3 Relationship between the searching efficiency of *N. californicus* and density of *T. urticae* under moderate to high temperatures.**

16, 20, 24, 25, 30, 32, and 40). With a type I functional response, *N. californicus* can consume more prey as the prey density increases, up to a certain point where it reaches its maximum handling capacity. At low prey densities, the predator cannot satiate its hunger and hence can consume all prey encountered.

In predator-prey interactions between *N. californicus* and *T. urticae*, the predator exerts significant control on prey populations at low *T. urticae* densities (4, 5, 8, and 10) because the predator can effectively control and reduce the number of prey when prey density is

low. However, as prey density increases beyond a certain point, the predator's predation rate saturates, reducing its efficiency in prey control. There exists an optimal prey density at which *N. californicus* is most effective in controlling *T. urticae* without reaching saturation. The population dynamics of *N. californicus* and *T. urticae* may show cyclic oscillations. When the population of *T. urticae* is low, *N. californicus* will thrive and multiply, which will cause the *T. urticae* population to fall. The growth of *N. californicus* will slow down as the *T. urticae* population becomes scarcer, which might give it time to recover. These interactions can impact ecosystems by benefiting plant health through reduced *T. urticae* infestation. Empirical studies and modelling are crucial to determine precise parameters and density thresholds for *N. californicus*'s type I functional response.

Our results showed that the attack rate and the handling time of *N. californicus* were influenced by different temperature levels. The attack rate was found to increase with increase in temperature. The maximum attack rate was recorded at 36 °C and maximum handling time at 24 °C. This demonstrated that as the temperature increased, the predation activity did as well. The reason behind such a response is that the predators spend more time in non-searching activities (*e.g.*, resting) at low temperatures, but they are more active in searching and predatory activity at higher temperatures. This can be explained by the fact that at high temperatures, predators are more active and have higher rates of reproduction (*De Clercq, 1993*), which causes them to consume more energy, as evidenced by the high predation rate. Handling time decreased, and the successful attack rate increased with an increase in temperature in a study with the observations every 24 h for seven consecutive days (*Rahman et al., 2012*) on the functional response of *N. longispinosus*, to the red spider mite, *Oligonychus coffeae*, where the prey densities used were 1, 5, 10, 15, 20, 25, and 30 adult females of *O. coffeae* with the range of temperature from 10 °C, 15 °C, 20 °C, 25 °C, 30 °C, and 35 °C.

Our findings support a study on pentatomids (*Podisus maculiventris* and *Stiretrus anchorago*) which found that the predation rate increased with increasing temperature (*Waddill & Shepard, 1975*). An earlier investigation, which chiefly focused on adult male and female, revealed a similar relationship between temperature on the predatory activity and functional response of the predatory thrips against the hawthorn spider mite (*Ding-Xu, Juan & Zuo-Rui, 2007*). Many other scientists (*Vasseur & McCann, 2005*; *Englund et al., 2011*; *Sentis, Hemptinne & Brodeur, 2012*) have also reported on the impacts of temperature on the functional response parameters causing striking modifications in the interactions between predators and prey, life histories, and food web linkages. It was described that temperature has a significant effect on the instantaneous attack rate and handling time of the predatory thrips since they spent more time, consuming pest mites at low temperatures and less time at high temperature (*Pakyari et al., 2009*).

Maximum predation (*K*) and *a/Th* values were recorded at the egg stage at 24 °C to 30 °C, followed by larval or nymphal stages in an irregular pattern, demonstrating that the predator consumes more prey as the temperature rises. Such results suggest that *N. californicus* shows good consumption-ability at higher temperatures up to 30 °C. This result confirms the finding (*Zamani et al., 2006*) that *N. californicus* can perform as a better candidate for the control of *T. urticae* in late spring and early fall inside greenhouses of the

countries with Mediterranean climate like Iran, where the temperature is frequently above 25 °C. As the density of the prey increased, so did the predator's rate of feeding. *Sandness & Mc Murtry (1970)* noted that *A. largoensis* exhibited a similar response, the reason behind which can be attributed to the stimulation–interference component, which resulted in the predator killing several preys in close succession because of increased prey contact. Another explanation according to *Sandness & Mc Murtry (1970)* for this behaviour is that at lower prey densities, the predator may begin to relax after feeding or laying an egg; however, at higher prey densities, the prey keeps bumping the predator and this bumping behaviour may cause the predator to attack the disturbing prey or to start looking for other preys.

Predators in the current study indicated a preference for eating larvae, followed by eggs, nymphs, and adults of *T. urticae*, which prevents the spider mite population from growing. Our findings are supported by a prior investigation on *N. longispinosus* where predators showed a stronger affinity for larvae compared to the nymphal stage of prey and a little preference for egg stage (*Rahman et al., 2013*). A study on the Japanese strain of *N. californicus* indicated greater agreement with our findings (*Canlas et al., 2006*) where predators preferred larvae to the eggs of *T. urticae*. Although the eggs and larvae of *T. urticae* provided the same type of nutrition to the predators (*Sabelis, 1985*), the preference for the larval stage may be because of the cluster feeding of larvae which makes their handling easy. The possible explanation for the less preference for the eggs over the larvae is the thickness of the egg chorion and the longer handling period. The difference in feeding preference can be attributed to the difference in the types of mouthparts of generalist and specialist predator; specialist predators have mouthparts that may puncture through the chorion of spider mite eggs, but generalist predators may not have such efficient mouthparts (*Blackwood, Schausberger & Croft, 2001*). In the present study, the predator's searching efficiency dropped as the prey density increased, which is consistent with the finding of *Zheng et al. (2017)*. The higher searching efficiency was recorded at the larval and egg stages. There is very little information in the literature regarding searching efficiency. Numerous variables, such as searching arenas, the impact of common predators (such as spiders and ants), spatial complexity, and weather, might negatively impact the effectiveness of natural enemies in the field (*Gitonga et al., 2002*).

Our study reveals that when *N. californicus* is released at its younger stages, including larvae and eggs, followed by nymphal stage, it can be a very effective biological control agent of the two-spotted spider mite, *T. urticae*. Additionally, the moderate to high temperatures that were examined in our study indicate that *N. californicus* can function well in warm environments other than typical field temperatures in both green houses and open fields. However, in order to ensure that the releasing of *N. californicus* at higher temperatures will be successful, additional field trials are required.

## CONCLUSION

From the current study, it can be concluded that the temperature has a significant impact on functional response and that *N. californicus* has significant predatory potential on *T. urticae* at higher temperatures. As the attack rate and effectiveness of predation

increased with increase in temperature, so did the prey consumption. These laboratory findings may be useful for further research on *T. urticae* management strategies and incorporating *N. californicus* in IPM programmes to effectively manage pest mites. The equations formulated in the present study can be utilized in IPM programmes to roughly evaluate the number of prey that is likely to be consumed by predatory mites at various temperature regimes, if the initial prey density is given.

### Funding
This study was funded by the National Key Research and Development Program of China (2018YFD0201400), and the Modern Agricultural Industry Technology System of the Sichuan Innovation Team (sccxtd-2019-04). The funders had no role in study design, data collection and analysis, decision to publish, or preparation of the manuscript.

### Grant Disclosures
The following grant information was disclosed by the authors:
National Key Research and Development Program of China: 2018YFD0201400.
Modern Agricultural Industry Technology System of the Sichuan Innovation Team: sccxtd-2019-04.

### Competing Interests
The authors declare that they have no competing interests.

### Author Contributions
- Maryam Mumtaz conceived and designed the experiments, performed the experiments, analyzed the data, prepared figures and/or tables, authored or reviewed drafts of the article, and approved the final draft.
- Vattakandy Jasin Rahman conceived and designed the experiments, analyzed the data, prepared figures and/or tables, authored or reviewed drafts of the article, and approved the final draft.
- Tahseen Saba analyzed the data, authored or reviewed drafts of the article, and approved the final draft.
- Tingting Huang analyzed the data, authored or reviewed drafts of the article, and approved the final draft.
- Yuxin Zhang analyzed the data, authored or reviewed drafts of the article, and approved the final draft.
- Chunxian Jiang analyzed the data, authored or reviewed drafts of the article, and approved the final draft.
- Qing Li conceived and designed the experiments, authored or reviewed drafts of the article, and approved the final draft.

## Data Availability

The raw data are available in the Supplemental File.

## Supplemental Information

Supplemental information for this article can be found online at http://dx.doi.org/10.7717/peerj.16461#supplemental-information.

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
