# Peer review of "Functional response of Neoseiulus californicus (Acari: Phytoseiidae) to Tetranychus urticae (Acari: Tetranychidae) at different temperatures"

_PeerJ, doi:10.7717/peerj.16461_

## Round 0.1 · original submission · Major Revisions

Dear Dr. Mumtaz and colleagues:

Thanks for submitting your manuscript to PeerJ. I have now received three independent reviews of your work, and as you will see, the reviewers raised some concerns about the research. Despite this, these reviewers are generally optimistic about your work and the potential impact it will have on research studying mite-on-mite biocontrol. Thus, I encourage you to revise your manuscript, accordingly, taking into account all of the concerns raised by the three reviewers.

All three reviewers provided many comments that, when addressed, will greatly improve your manuscript. Please check over the English and writing. There seem to be some improvements that can be made. Also, strive for consistency with symbols and abbreviations. There appears to be some missing information regarding methodology. Please include all information to make your work repeatable and all data accessible.

Finally, be careful about interpretations of results (for IPM, greenhouse versus field).

Please note that reviewer 3 kindly provided a marked-up version of your manuscript.

I look forward to seeing your revision, and thanks again for submitting your work to PeerJ.

Good luck with your revision,

Best,

-joe

Reviewer 1 ·

Basic reporting

The authors conducted a study to examine the relationship between predator (Neoseiulus californicus) consumption rate and prey (Tetranychus urticae) density at different temperatures. This research provides valuable insights into predator-prey interactions and contributes to the development of Neoseiulus californicus as a biological control agent for spider mites. Overall, the paper is well-written. The introduction provides sufficient background information to understand the study, and the paper's structure is acceptable. However, I have a few suggestions for improvement:
• Throughout the paper, the notation for the time required by the predator to handle the prey (Th) is inconsistent. For instance, 'h' is used as a subscript in Line 131 but not in Line 152, and 'Th' is italicized in Line 133 but not in Line 131. Please ensure consistency in notation throughout the paper.
• In Line 131, I recommend checking the Holling disc equation and consider adding parentheses before and after '1+aThN'.
• Table 1 requires further editing. For example, in column 'Na', some equations are in font size 11 while others are in size 10. Additionally, although parentheses were added for '1+aThN', they are missing in some equations. Please ensure consistent formatting throughout the table.
• In Line 156, please include the statistical method used to determine the significance of 'a/Th' values between larval and egg stages.

Experimental design

The authors should provide a more detailed description of their experimental methodology. Specifically, it would be helpful to know:
• Where was the experiment conducted? Was it in an incubator, greenhouse, or in the field?
• How was the temperature monitored throughout the experiment?
• Were different stages of spider mites added to the same plant?
• How did you ensure that mites did not move from one plant to another during the experiment?

Validity of the findings

no comment

Reviewer 2 ·

Basic reporting

Professional English is used throughout. However, some sentences should be rewritten to gain clarity. For example, L. 44 “huge trees”, L. 152 “was found maximum… “, L. 154 “the handling time values generally altered”, L. 254 “N. californicus is released at younger stages…” – of T. urticae? “function well in warm environments other than typical field temperatures” – for release in greenhouses? Punctuation should be checked throughout the manuscript as well as the structure of some paragraphs (for example, the 3rd paragraph of the introduction).

The article includes an extensive list of references on the use of predatory mites as biological control agents with > 10 references on T. urticae as prey and N. californicus as predator.

Some references are not complete (Copping LG 2001, De Clercq P 2003, Helle W, Sabelis MW 1985), and some do not seem the most relevant when cited (for example, L. 43 Pavela 2017, L. 49 Wu et al. 216, L. 52 Uddin et al 2017).

The structure of the article follows the format of “standard sections”. Figures and Tables are relevant and appropriately described and labeled, except in Figure 3 where there is a spelling mistake in T. urticae and twice the same x-axis legend. It is not the case in the raw data file. Furthermore, the raw data file gives for each treatment 5 replicates and not 3 as indicated in the manuscript. Legend of Figure 2 should be “mean number of preys consumed - or mean consumption rate”. In Figure 1, would it be possible to give SE as means are indicated in the legend of the y axes?

Statistical analyses are not shown or are lacking.

Experimental design

The work focuses on the functional response of N. californicus. How it is expected to be similar or different from functional responses calculated for other predatory mites (Fantinou et al. 2012 etc.), or for N. californicus (Ahn et al. 2010, Farazmand et al. 2012 etc.) is not clearly stated. Are the densities of T. urticae and/or the range of temperatures in the experimental design similar to previous studies?

Methods are not described with sufficient information to be reproducible by another investigator: how were the different stages of T. urticae supplied to the females of N. californicus ? How many N. californicus females were used for each treatment ? Were the different treatments (density × temperature) conducted simultaneously? L. 128 How was the “searching efficacy” evaluated?

Are the differences among stages and/or temperatures significant for the different parameters that were calculated ?

Validity of the findings

The article includes an extensive list of references but does not clearly demonstrate how the work improves our understanding of predator-prey interactions and the dynamics of food webs. In particular, the introduction does not state how the study is expected to bring new insights into the ecology of predatory mites in general, and of N. californicus in particular, nor how it compares to other experiments on the biological control of T. urticae.

Most of the results and discussion focus on the quantification of the functional response of N. californicus. Type II functional responses seem to prevail in nature (Jeschke et al. 2004) and correspond to a density-dependent predation with a decelerating consumption rate as prey density increases. L. 160 it is stated that “it was fitted to the Holling Type II model”. Figure 2 should therefore show how the data fit a curvilinear curve for the different stages of T. urticae. As shown, the relationships between prey density and consumption rate seem to be linear at all stages.

Some questions are left unanswered:

- The authors strongly state that at higher temperatures, predation rate by N. californicus is also higher. However, there are differences among stages with a (significant?) drop in predation (K and a/Th) of eggs at 33°C and 36°C, and of nymphs at 36 °C. How could this result be explained and what would be the consequences when using N. californicus as biological control agent?

- Why is digestion time included in the estimate of handling time? How was it considered? See recent work by Papanikolaou and colleagues (2020, 2021 etc.) for discussion on this question.

- Is it possible to discuss the impact of the size of the data set on the estimates of the different parameters? See for example recent work by Papanikolaou and colleagues (2020, 2021 etc.).

- What are the data on the ecology of N. californicus which could support the discussion on why feeding rate is higher at higher prey density ? There are no references for arguments such as “the interference caused by the predator to the normal activity of the predator” or “the predator may begin to relax after feeding and laying an egg” etc.

In the conclusion, caution should be made to extrapolate results from a laboratory experiment to IPM in a greenhouse or a field setting. Other parameters might influence the functional response of N. californicus.

Reviewer 3 ·

Basic reporting

In general, the article provides good information which can be used in Biological control programmes when using N. californicus. However, the English language should be reviewed and the suggested corrections incorporated.
In terms of the literature reference used, when possible, some of the oldest ones should be replaced with a more recent reference on the topic.
The structure of the article is aligned with the requirements! The content is clear! Some suggestions were done in terms of how the results following the tables and the figures were presented in the article. A few sections are confusing!

Experimental design

The experimental design and presentation of the results were well aligned with the objectives and questions definition. The analytical tools used were considered appropriate!

Validity of the findings

In general, the article provides good information which can be used in Biological control programmes when using N. californicus. However, not really novel information was provided.
The data provided and presented in the results was well analyzed and statistically sound. The conclusions were clear and aligned with the objectives of the study!

Additional comments

Despite the article does not provide novel information, the results could contribute to better use of N. californicus in the tropical areas with higher temperatures!

Annotated reviews are not available for download in order to protect the identity of reviewers who chose to remain anonymous.

---

## Round 0.2 · Minor Revisions

Dear Dr. Mumtaz and colleagues:

Thanks for revising your manuscript. The reviewers are mostly satisfied with your revision (as am I). Great! However, there are a few remaining concerns to address (per reviewer 2).

Please address these ASAP so we may move towards acceptance of your work.

Best,

-joe

Reviewer 1 ·

Basic reporting

I am pleased to note that the concerns I raised have been successfully addressed in the revised manuscript.

Experimental design

I am pleased to note that the concerns I raised have been successfully addressed in the revised manuscript.

Validity of the findings

No comments.

Reviewer 2 ·

Basic reporting

Basic reporting has been improved and examples of unclear sentences have been rewritten. Some might still need some editing.
Literature references have been edited and updated when possible. Some references have been added to support suggested revisions.
Mistakes on the number of replications, figure legends and axes, lack of error bars etc. have been corrected.

Experimental design

The experimental design has been clarified.
The work focuses on the functional response of N. californicus at different temperatures and prey densities (X stages). The authors expected a Type II functional response. Is it possible to compare the range of temperatures or densities of T. urticae with previous studies? This point does not seem to have been answered.

Validity of the findings

In terms of significance of the work, the authors mention that the results will help to design of IPM strategies with N. californicus as biological control agent. However, the results after revision indicate a Type II response to T. urticae adults only, and Type I to larvae and nymphs. How is it possible to explain this result (density range? behavior? etc.). What are the consequences for our understanding of predator-prey dynamics? This point is not really discussed in the revised manuscript.
Similarly, a few other points should be addressed :
- For the discussion on handling time that includes or not digestion time, would it be possible to justify why the consumption rate was estimated after 24 h?
- How does the size of the data set compare to previous studies?
- Is there any other (more recent) references than Sandness and McMurtry (1970) on the relationship between feeding rate at different prey densities?

---

## Round 0.3 · accepted · Accept

Dear Dr. Mumtaz and colleagues:

Thanks for revising your manuscript based on the concerns raised by the reviewers. I now believe that your manuscript is suitable for publication. Congratulations! I look forward to seeing this work in print, and I anticipate it being an important resource for research on mite-on-mite biocontrol. Thanks again for choosing PeerJ to publish such important work.

Best,

-joe